

# Open-source data management system for Parkinson's disease follow-up

João Paulo Folador[1], Marcus Fraga Vieira[2], Adriano Alves Pereira[1] and Adriano de Oliveira Andrade[1]

[1] Centre for Innovation and Technology Assessment in Health, Postgraduate Program in Electrical and Biomedical Engineering, Faculty of Electrical Engineering, Federal University of Uberlândia, Uberlândia, Minas Gerais, Brazil

[2] Bioengineering and Biomechanics Laboratory, Federal University of Goiás, Goiânia, Goiás, Brazil

## ABSTRACT

**Background:** Parkinson's disease (PD) is a neurodegenerative condition of the central nervous system that causes motor and non-motor dysfunctions. The disease affects 1% of the world population over 60 years and remains cureless. Knowledge and monitoring of PD are essential to provide better living conditions for patients. Thus, diagnostic exams and monitoring of the disease can generate a large amount of data from a given patient. This study proposes the development and usability evaluation of an integrated system, which can be used in clinical and research settings to manage biomedical data collected from PD patients.

**Methods:** A system, so-called Sistema Integrado de Dados Biomédicos (SIDABI) (Integrated Biomedical Data System), was designed following the model-view-controller (MVC) standard. A modularized architecture was created in which all the other modules are connected to a central security module. Thirty-six examiners evaluated the system usability through the System Usability Scale (SUS). The agreement between examiners was measured by Kendall's coefficient with a significance level of 1%.

**Results:** The free and open-source web-based system was implemented using modularized and responsive methods to adapt the system features on multiple platforms. The mean SUS score was 82.99 ± 13.97 points. The overall agreement was 70.2%, as measured by Kendall's coefficient ($p < 0.001$).

**Conclusion:** According to the SUS scores, the developed system has good usability. The system proposed here can help researchers to organize and share information, avoiding data loss and fragmentation. Furthermore, it can help in the follow-up of PD patients, in the training of professionals involved in the treatment of the disorder, and in studies that aim to find hidden correlations in data.

# INTRODUCTION

Parkinson's disease (PD) is a neurodegenerative condition of the central nervous system that affects the basal nuclei, because of a progressive loss of dopaminergic neurons of the substantia nigra. The decreasing of these neurotransmitters causes motor and non-motor dysfunctions, postural and cognitive disorders (*Andrade et al., 2013*).

Corresponding author
João Paulo Folador,
jpfolador@gmail.com

Parkinson's disease affects about 1% of the world population of individuals over 60 years old, and it is among the five most expensive neurological disorders in European Countries. The Brazilian National Health System spends an average of US$ 5,853.50 with an annual cost of PD per patient, including 53.4% direct costs and 46.6% indirect costs (*Bovolenta et al., 2017*). The PD diagnosis is not simple; however, symptoms such as tremor, stiffness, bradykinesia, and postural instability are cardinal signs of the presence of the disease. Monitoring the symptoms that affect the individual with PD can bring better living conditions to the patients since the disease remains cureless (*Andrade et al., 2013*; *Kassavetis et al., 2016*).

The evaluation of PD symptoms is commonly performed using clinical scales such as the Movement Disorder Society–Unified Parkinson Disease Rating Scale (MDS–UPDRS) and Parkinson's Disease Questionnaire (PDQ-39). Several exams are also used to further assess the condition of the patient over time, for instance, the electrocardiogram, electromyogram, electroencephalogram and medical imaging exams (e.g., Magnetic Resonance Imaging). Recently, inertial data, measured by inertial sensors, have been widely used to study the motor condition of patients (*Andrade et al., 2013*).

As a consequence of the technological advancement in PD evaluation, there has been an increase in the volume and types of available data. In this sense, the organization and management of data are essential to obtain useful information from a large amount of data (*Dinov et al., 2016*). Consequently, it is necessary to know the technologies used to optimize data collection and storage (*Roibu Crucianu, 2019*). This amount of generated data presents new challenges in the areas of information management, storage, security, and difficulty in understanding the produced data (*Dinov et al., 2016*; *Roibu Crucianu, 2019*; *Agrawal & Prabakaran, 2020*).

The multimodality and complexity of data concerning PD evaluation have been increasing, as shown in several studies. For instance, *Klinger et al. (2006)* developed a virtual environment that simulates a supermarket on the personal computer (PC) through which the volunteer performs planned tasks to purchase goods. The authors recorded the trajectory of the patient in the virtual environment to assess the cognitive deficit of task planning.

*Cunningham et al. (2009)* implemented a desktop software to assess the level of hand and finger control in PD patients. The participants performed the tasks of clicking buttons that alternated on the software screen. The authors collected and stored data such as personal information, speed of clicks, the coordinates, and the trajectory of the movement to evaluate the rigidity and dexterity.

*Pastorino et al. (2013)* recorded inertial data from the upper and lower limbs and the waist of people with PD in the ON and OFF periods of medication. Similarly, *Caldara et al. (2014)* proposed a network of wireless inertial sensors attached to the limbs and body to assess the gait, posture, and tremor of individuals with PD. The desktop software captured data via Bluetooth and saved information in a text file. The cadence, step length, and stride length of PD patients were recorded by *Arango Paredes et al. (2015)* through a Kinect, a motion-sensing technology based on RGB cameras, infrared projectors, and detectors.

*Eskofier et al. (2016)* used wearable inertial sensors in PD patients to detect bradykinesia. The authors used Deep Learning on the sensor data and reported an accuracy of 90.9% in the classification of the individuals. *Kassavetis et al. (2016)* evaluated the symptom of bradykinesia and tremor in PD patients using software developed for mobile devices, which employs the capacitive screen and the accelerometer to collect and store the data through a smartphone.

The investigation of speech problems in people with PD was proposed by *Dimauro et al. (2017)*, who used the Google Speech-To-Text tool to assess speech intelligibility. The authors used reading exercises to perform the speech evaluation to assist specialists in improving PD treatments. In another study, *Haddock et al. (2018)* proposed a tool to control the parameters of deep brain stimulation devices automatically. The authors used inertial sensors from a Smartwatch that captures, via Bluetooth, the tremor on the most affected hand.

Specific systems developed to manage people with PD and other neurological disorders have been previously reported. For example, *Cancela et al. (2013)* showed a way of integrating interinstitutional databases focused on studying medical images. The system uses a server, an application programing interface, and a query builder to ease the exchange of information between researchers and their databases.

*Pepa et al. (2015)* developed an application on the smartphone to monitor the gait of patients with PD. The data were stored periodically on the device and synchronized with web-based software. Similarly, *Patel et al. (2010)* monitored patients at home using wearable sensors that transmitted the data via a mobile device to a web application. *Garzo et al. (2018)* developed a web-based system to monitor and record gait and freezing episodes in people with PD. The type of monitoring (*Pepa et al., 2015*; *Patel et al., 2010*; *Garzo et al., 2018*) can produce a large amount of data.

The development of any system for the management of biomedical data should be followed by evaluating its usability, which aims to understand whether a such system is easy to use and has the appropriate functionality for the users. A method commonly employed to assess usability is the System Usability Scale (SUS), which has been applied in several situations such as the evaluation of a mobile application that helps to improve gait in people with PD (*Garzo et al., 2018*), an augmented reality game to evaluate the extremity of upper limb impairment in stroke (*Bank et al., 2018*) and PD patients (*Bank et al., 2018*; *Van der Meulen et al., 2016*), evaluating a mobile application for mental health monitoring that collect physiological signals, activity, and environmental data (*Kamdar & Wu, 2016*), testing the usability of augmented reality software in food advertising via smartphone (*Wijaya, Munandar & Utaminingrum, 2019*), learning evaluation of a management web system (*Katsanos, Tselios & Xenos, 2012*), evaluation of a multimedia interface for learning English (*Devy, Wibirama & Santosa, 2018*), and also to test e-commerce application on smartphones (*Indriana & Adzani, 2017*).

In this scenario, the amount of collected data may have a large volume, increases fast and assumes a variety of data formats (*Dinov et al., 2016*). Furthermore, if these data are not organized and secure, being in any place such as clinics, hospitals, and research centers, the information can be lost, fragmented, poorly analyzed, and the resources

invested are wasted. In addition, it is crucial controlling of sensitive personal information and guarantee the privacy rights formalized as in California Consumer Privacy Act (*Goldman, 2018*), Data Protection Law Enforcement Directive of European Union (*Quintel, 2018*) and General Personal Data Protection Act in Brazil (*Da Silva, Da Luz Scherf & Da Silva, 2020*).

From the literature review, it is possible to identify the lack of systems capable of managing biomedical data related to the knowledge and follow-up of PD patients. Besides that, many philanthropic institutions and PD associations that help people with PD in countries such as Brazil have limited technical resources to manage their basics services. In this context, it is of utmost importance to develop free and open-source systems.

This research presents different aspects to the management of information of people with Parkinson's disease: (1) development of a system that integrates information and research files, (2) assessment and clinical monitoring through customizable scales and questionnaires, (3) learning and training about Parkinson's disease, (4) control at distinct levels of security in a modularized and multiplatform format, (5) the usability test (SUS) of the system performed by 36 examiners, and (6) a free and open-source initiative to assist, mainly, philanthropic institutions and PD associations that help people with PD.

## MATERIALS AND METHODS

### Requirement analysis

The development of the system was based on the Rapid Application Development (RAD) concept. RAD is an approach based on agile project, and it was chosen because this methodology is incremental, emphasizing rapid and reusable coding for the development of application modules. It can be understood in four main steps: (i) requirement planning: developers, software users, and team members discuss the objectives and expectations of the project and address the requirements; (ii) design: in this step, a prototype is drawn up and discussed with software users and refined until an acceptable design is reached; (iii) construction: this step is mainly related to database development and coding of the prototypes. (iv) cutover: final coding, tests on the system, and the system users are trained. Bugs can be reported for correction, and new requirements will reinitialize step (i). This concept focuses on lean documentation, relying on the essentials for coding the functionality of the system, but requires an experienced programmer (*Jailia et al., 2016*; *Sommerville, 2010*; *Qodim, Busro & Rahim, 2019*).

In line with the RAD concept, several functionalities were analyzed to construct an integrated system providing an adequate tool. To reach this objective, the process was guided by the expertise of seasoned professionals in research on PD, interviews with specialists in PD diagnosis, and professionals specialized in software development. Furthermore, the development of the system took into consideration all aspects and conditions of PD patients (biological, social and cultural), and a complete study of well-known questionnaires and clinical scales to evaluate PD patients (*Andrade et al., 2013*).

All the identified requirements were used to develop the system architecture and organization, to design the graphical user interface, and to model data structure. The

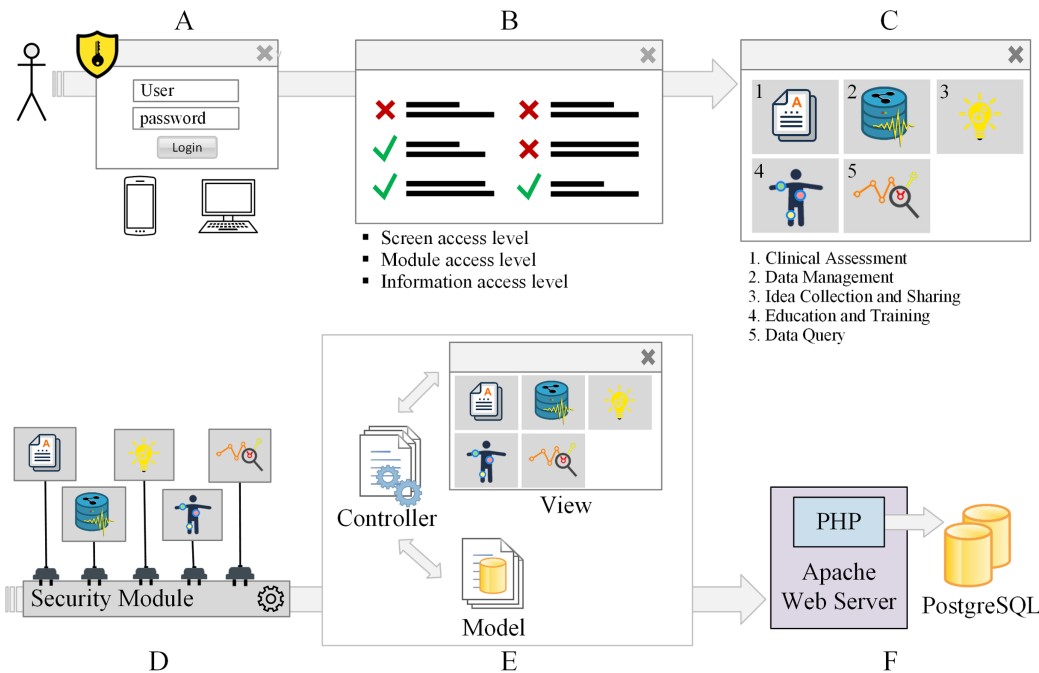

**Figure 1 Access, security, and system structure flowchart.** The system requires user authentication (A) and user profile verification (B) to access the modules (C). The modules of the system are connected to a security module (D). The system architecture is based on the model-view-controller pattern (E). The system is hosted on the Apache webserver, and the data are stored in the PostgreSQL database (F).

system was developed incrementally, that is, it was tested and improved as the unit was developed (*Sommerville, 2010*).

## Integrated biomedical data system architecture

The aim of the Integrated Biomedical Data System (in Portuguese, Sistema Integrado de Dados Biomédicos (SIDABI)) is to manage and secure data from PD patients and research volunteers. Fig. 1 depicts the architecture and structure of the system.

Figure 1 shows the security levels that a system user or administrator has to go through to access the distinct modules of the system. Fig. 1A illustrates the first security barrier to authentication. The access to specific screens and modules is released according to the user's profile (Fig. 1B). Users may have permission to create, view, update and delete records. Each module of the system is presented in Fig. 1C. The system has five interconnected modules that can share information among them: Clinical Assessment, Data Management, Idea Collection and Sharing, Education and Training, Data Query. All the modules of the system are connected to a security module, Fig. 1D.

The system architecture is based on the Model-View-Controller (MVC) pattern (Fig. 1E). The Apache web server was used to host the system because it is the most used web server. Apache web server is an open source project, easy to customize environments, fast, reliable, and highly secure (*Fielding & Kaiser, 1997*; *Baş Seyyar, Çatak & Gül, 2018*) and the Apache web server can be set up to accept distinct programing languages and databases. The data are stored in the PostgreSQL database that also is open-source

software, secure, robust, works with relational and no relational data, highly concurrent, and the most popular database system (*Hellerstein, 2018*). The Apache web server and the database are illustrated in Fig. 1F

## Design pattern

In general, the design pattern provides best practices in a framework of solutions for common problems in the context of designing software. The MVC is a pattern used to build interfaces in three separated logic layers: the Model manages the data and rules of the application; the View is the representation and appearance of the data on the screen such as text, tables, charts; the Controller handles the requests from View or Model layers validating and processing these data, and sending commands to execute the appropriate action (e.g., store data, retrieve data, show some information in a table) (*Jailia et al., 2016*; *Sommerville, 2010*; *Syromiatnikov & Weyns, 2014*).

In the literature, it is possible to find recent patterns based on MVC such as Model-View-ViewModel (MVVM) in which the ViewModel layer controls the functions of presentation, sending commands, automating communication between View and ViewModel layer, and supporting the View state. MVVM supports multiple views. It was created to facilitate interactions between the View and ViewModel layers of graphical user interfaces. The Model-View-Presenter (MVP) is more flexible about layer's responsibilities, and the pattern also was based on MVC. The Presenter layer is responsible for supervising the View and Model layers, handling user events, listening to the View layer, making changes to the View layer, and maintaining the application synchronized. In general, the Model and View layers in MVVM and MVP follow the principles of the MVC pattern (*Syromiatnikov & Weyns, 2014*).

Despite some improvements in MVVM and MVP, the MVC design pattern was used to construct the proposed web system because it is the most acceptable standard used to develop custom applications on the Web in different programing languages (*Jailia et al., 2016*; *Sommerville, 2010*). Additionally, MVC gives more focus on the Controller layer (*Syromiatnikov & Weyns, 2014*) that is suitable for this study once the main idea is based on controlling requests made by the software user and verifying its permissions of access in a module, page and data manipulation. These verifications can slow the response of requests in some milliseconds, MVC can result in additional classes and objects, and in more complex file organization, however, the MVC pattern also allows reliable access to the database, providing a clean, organized, reusable, scalable, and efficient code (*Jailia et al., 2016*; *Sommerville, 2010*).

The MVC pattern is used by a number of well-known frameworks and companies such as Django (National Geographic, Instagram, Pinterest, Open Stack, Mozilla) (*Django Web Framework, 2021*), Ruby on Rails (GitHub, Airbnb, Basecamp, Shopify) (*Hansson, 2021*), Spring (Netflix, VMware Tanzu, Alibaba, Amazon, Google, Microsoft) (*Spring Framework, 2021*). Despite this, software that uses the MVC pattern in the medical field related to Parkinson's disease is still rare. Studies found in the literature showing software development, such as the use of a mobile and web application to improve gait in patients with PD (*Garzo et al., 2018*), an augmented reality game to assess upper limb impairment

in stroke (*Bank et al., 2018*) and PD patients (*Bank et al., 2018*; *Van der Meulen et al., 2016*), and a mobile mental health monitoring application (*Kamdar & Wu, 2016*), did not describe any pattern of development used.

Figure 1E illustrates the MVC adopted, in which the interaction between the user and the system's response to the user is represented. The View layer contains the elements presenting information to the software user, and these elements can be built using HTML5 (the language used to create the structure of the elements shown on web pages), Cascading Style Sheets (CSS3), which is the mechanism for formatting the pages, and JavaScript (JS) that is the programing language used on the client-side to make it possible the interaction between the user and the components presented on the screen (*Jailia et al., 2016*; *Freeman et al., 2004*). The JS library JQuery 3.1.1 was used to optimize interactions.

Therefore, when the user interacts with the system, for example, by clicking on a button, JS handles this event and passes it on to a controller. The controller layer uses the PHP programing language to code the server-side system. The view controller assembles the results brought by the model controller or simply assembles the response of a specific request and returns it to the browser. The model controller is responsible for interacting with the model layer, which has classes that interact directly with the database (*Jailia et al., 2016*; *Majeed & Rauf, 2018*).

In addition to the MVC pattern, the Responsive Web Design (RWD) concept was also used to design the layout with optimized experience, good ergonomics and usability while using the system on different devices. The Bootstrap Framework 3.3.7 was used to ensure proper functioning on different platforms (*Majeed & Rauf, 2018*).

## Usability evaluation

Usability is a term used to define how easy people can employ a tool or object in order to accomplish a specific task. Usability is also related to the evaluation of a system with the aim to improve (i) human computer interaction, and (ii) the social and practical acceptance of the system (*Nielsen, 2004*; *Ganney, Pisharody & Claridge, 2013*). Thus, the usability of an interface should be good enough allowing the users to perform the system's tasks easily.

In this way, the usability of a system should focus on developing interfaces that are easy to handle and quick to learn. The functionality of the layout should avoid and deal with operational errors efficiently and with appropriate feedback to the user. In addition, usability must address user satisfaction and provide an effective solution to the problem that the system was designed to solve (*Nielsen, 2004*; *Ganney, Pisharody & Claridge, 2013*).

In this sense, SUS was used to evaluate the usability of the system. The scale has ten questions, $q = \{q_1, \cdots, q_{10}\}$, and each of them can assume a value, $w$, ranging from 1 (strongly disagreement) to 5 (strongly agreement) (Table 1). The calculation of the scores for a specific question, $S_q$, is given in Eq. (1). The final scale score, $SUS_{score}$, ranges from 0 to 100 points, with 68 points being an acceptable score (*Jordan et al., 1996*), and this final score is given in Eq. (2).

**Table 1 SUS questions.**

| Question | Description |
|---|---|
| 1 | I think that I would like to use this system frequently |
| 2 | I found the system unnecessarily complex |
| 3 | I thought the system was easy to use |
| 4 | I think that I would need the support of a technical person to be able to use this system |
| 5 | I found the various functions in this system were well-integrated |
| 6 | I thought there was too much inconsistency in this system |
| 7 | I would imagine that most people would learn to use this system very quickly |
| 8 | I found the system very cumbersome to use |
| 9 | I felt very confident using the system |
| 10 | I needed to learn a lot of things before I could get going with this system |

$$S_q = \begin{cases} w - 1, & \text{if } q \text{ is odd} \\ 5 - w, & \text{if } q \text{ is even} \end{cases} \tag{1}$$

$$SUS_{score} = \left( \sum_{q=1}^{10} S_q \right) * 2.5 \tag{2}$$

*Bangor, Kortum & Miller (2009)* proposed an adjective rating scale according to the average points obtained from SUS, with the usability classified as worst imaginable (12.5), awful (20.3), poor (35.7), ok (50.9), good (71.4), excellent (85.5) and best imaginable (90.9).

The usability test (SUS) of the system was performed by 36 examiners, and the resultant data is available in the Supplemental Information section. The data is organized in a comma-separated file (usability-db-en.csv) in which the first four columns are related to the characterization of the examiners, and the fifth to fourteenth columns deal with the SUS questions.

In addition, Kendall's coefficient of concordance was estimated to verify the overall agreement between the examiners in the SUS questionnaire. It was used because it is a non-parametrical statistic, more accurate to smaller sample sizes, which can be used for assessing agreement among examiners considering ordinal data, and can be used for assessing agreement among a variety of raters. This coefficient can assume values from 0 (without agreement) to 1 (complete agreement) (*Legendre, 2005*).

The statistical analysis and data visualization were performed in R, a language and environment for statistical computing (*R Core Team, 2019*), using the open-source integrated development environment (IDE) RStudio for R, version 1.2.5042. Thus, an R file with the analysis presented in this study (*data-analysis-usability-v01.R*) is also available in the online repository pointed at the Supplemental Information section.

## Experimental protocol

This research follows the Resolution 466/2012 of the National Health Council. The study was conducted at the Centre for Innovation and Technology Assessment in Health of the

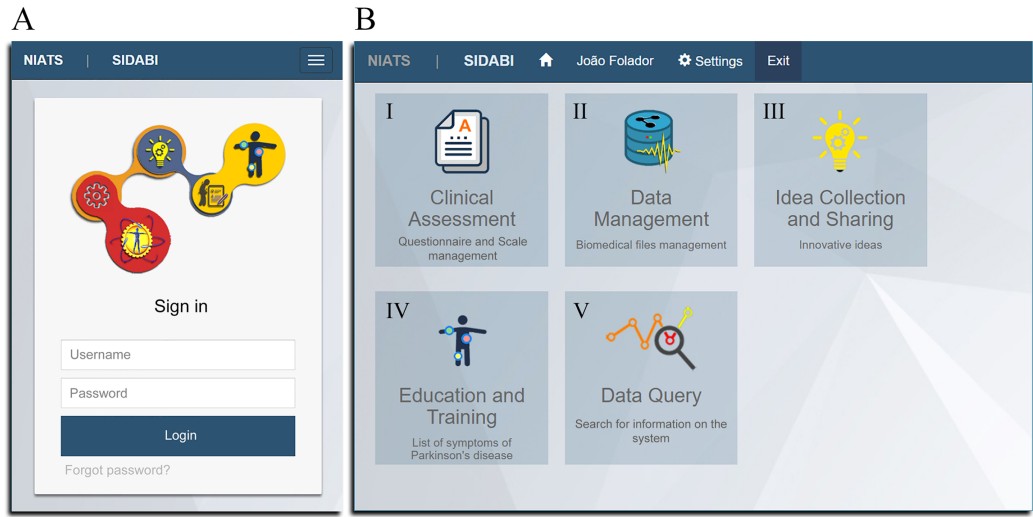

**Figure 2 Main graphical interfaces of the system.** (A) The viewing on mobile device of the authentication system screen. (B) The viewing on PC of the screen showing the modules of the system (I–V).

Federal University of Uberlândia (UFU), Brazil. The experimental protocol was approved by the Human Research Ethics Committee (CEP-UFU), CAAE Number: 93993118.4.0000.5152. The participants were informed about the data collection procedures and signed a consent form before data collection.

During the experiment, the system was evaluated by 36 examiners and the examiners were not trained on how to use the system before carrying out the proposed tasks. They remained comfortably seated and accessed the web system through a computer. In order to evaluate the usability of the system, the user was asked to execute several common tasks, such as logging in to the user account, accessing specific software modules and filling in form data, uploading experimental session data, using search fields, and others. The duration of the interaction tasks was approximately 15 minutes.

The user was asked to inform the level of computer skills (1–2, 3–5, more than 5 years), age (18–25, 26–35, 36–45, more than 46 years old), and the number of hours of computer use per week (2–5, 6–10, over 10 h). Subsequently, the examiner answered ten questions about the usability evaluation questionnaire.

## RESULTS

### System description and visualization

Figure 2 illustrates the authentication screen (A) and interface that provides access to the distinct modules of the system (B). The ordinary user or administrator performs authentication using a username and password. The security module validates access into screens, modules, and information manipulation levels that the user has on each page, such as inserting new data, data visualization, updating information, or data removal.

Currently, the system contains five implemented modules (Fig. 2B) and a security administration area to manage user permissions. Each module has its functionalities, and

some of them share data. The principal functions of the modules are described in Fig. 3 as a use case diagram. Besides that, the complete system can be downloaded and is available in the Supplemental Information section.

The use case diagram is a feature of Unified Modeling Language (UML) and is commonly used to show the relationship between functionalities and system users. It is used for a high-level view of the system and identifies the fundamental factors. The use case diagram utilizes some notations such as an actor that performs a role (i.e., a user and an administrator of the system), a system user can be researcher's labs, hospital staff, nonprofit organizations staff; an ellipse drawing that is the use case representing a function or an action in the system. The system object is represented as a rectangle, which defines the scope of the use case (*Sommerville, 2010*).

Besides that, as shown in Fig. 3, the relationships in a use case diagram are used to represent the interactions between actors and use cases. A continuous line connects an actor to functionality, a dashed line with an arrow and marked with an <<include>> word means a necessity to add that use case (the arrow points to the use case included). On the other hand, a dashed line with a <<extend>> label means a possible additional functionality but not mandatory (the arrow points to the use case that extends the functionality) (*Sommerville, 2010*).

In this context, Fig. 3 represents the main use case diagrams of SIDABI modules shown in this study. It is essential to emphasize the words *manage* and *control* presented in the use cases meaning junction of some data manipulation functionalities such as insert, edit, delete, and find information. Thus, Fig. 3A represents the security module with two actors, the administrator can manage all the permissions functionalities, and the simple user can visualize its account and the modules permitted. In Fig. 3B, the clinical assessment software shows two distinct roles, that is, the researcher, which is a person who can manage the main functions of the clinical assessment module and can apply a questionnaire, and the patient or caregiver role, that is the actor that can respond the questionnaire.

In Fig. 3C, the data management module is represented, and the system user with appropriated permissions can control the functionalities and upload the files resulting from the session of data collection. Fig. 3D illustrates the module so-called idea collection and sharing, which allows the system user to control new project proposals.

Figure 3E represents the education and training module from which the user can manage the information of the symptoms of PD and learn from a page that overviews the symptoms, tutorials, and extra materials about the disease. Fig. 3F shows the patient or volunteer management functions that are used to add and control the records of people in the system. Finally, Fig. 3G illustrates the data query module with an actor accessing a menu with filters that can help to narrow the results of research data. The software users can select the filters (e.g., study group, participant, equipment) to help them to get more accurate results, for instance, if it was chosen a specific equipment, the query result brings all the files related to that equipment. These functionalities mentioned above are detailed in Table 2. Additionally, the whole database diagram of SIDABI is available in the Supplemental Information section.

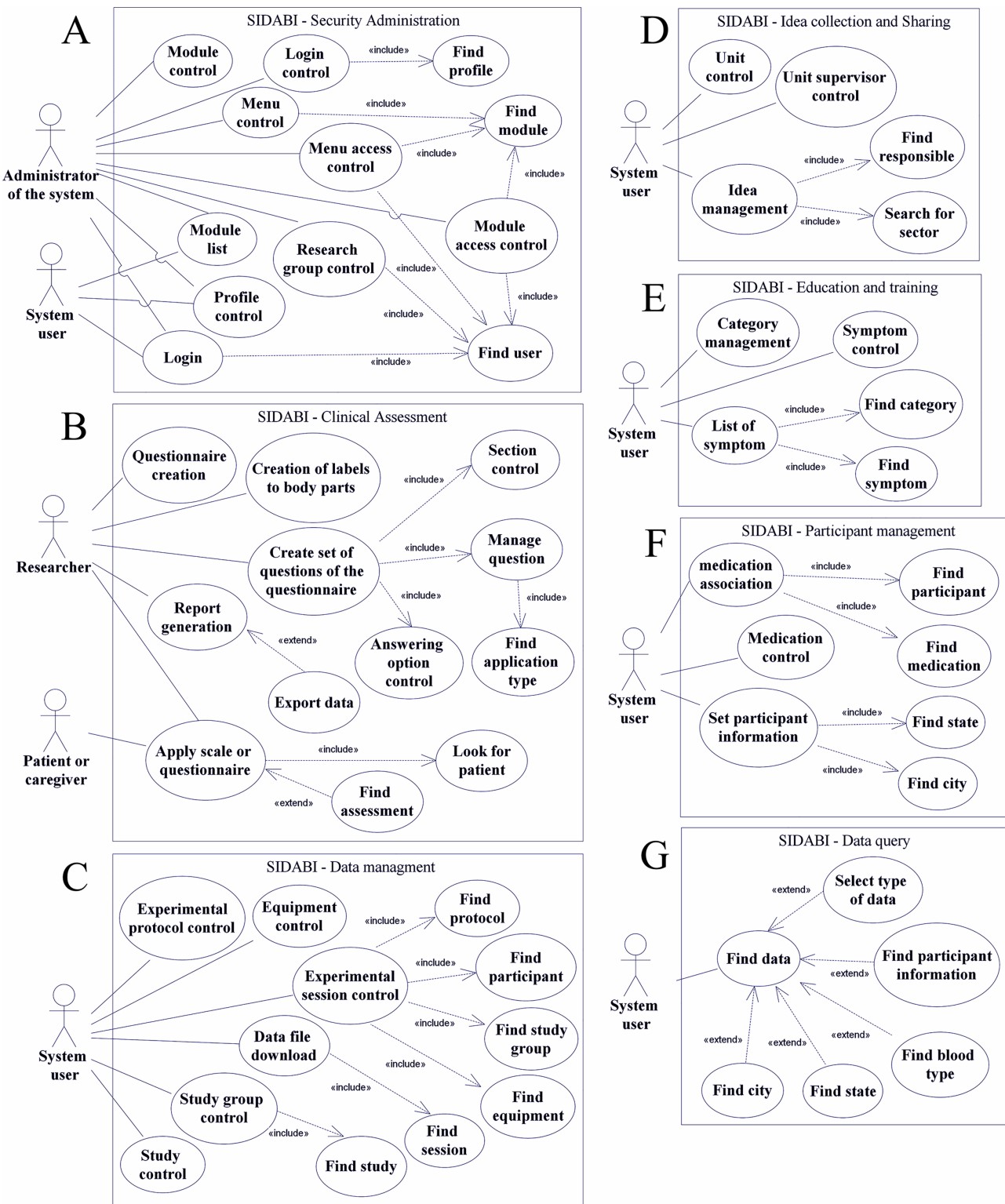

**Figure 3 Use case diagram of the main functionalities of SIDABI.** (A) Represents the security module, (B) the clinical assessment software, (C) the data management module, (D) the idea collection and sharing functionalities, (E) the education and training module, (F) the management of participants and (G) the data query module.

**Table 2 Main functionalities of SIDABI.**

| Module | Functionality | Description |
|---|---|---|
| Security Administration (settings) | Login control | This functionality allows registering a new user in the system. Specific information, such as name, username, password, e-mail, profile, and user credentials (i.e., administrator or common user) are required. |
| | Profile control | This sets the category of the user (e.g., student, professor, guest). |
| | Module control | This functionality controls the creation and exclusion of modules in the system. The user has to inform an acronym, title and image that represents the purpose of the module. |
| | Module access control | It is used to control the access of users to specific modules. |
| | Menu control | It allows the assignment of specific menus to the modules of the system. |
| | Menu access control | This functionality restricts the access of specific menus to specific users. |
| | Research group control | It is used to create research groups. |
| | Researcher association | This functionality allows associating a user and a research group. |
| Clinical Assessment | Questionnaire creation | It is used to create a customized questionnaire that may be a clinical scale. |
| | Creation of labels to body parts | In Parkinson's disease, it is very common to assess body parts through specific questions. The creation of identification labels for body parts (e.g., jaw, upper left limb, and right hand) allows for the recording of specific clinical evaluations. |
| | Create set of questions of the questionnaire | The user can create the whole questionnaire or scale with this functionality. It is possible to create question sections, questions, answering options. Each question can be linked to a label of a body part. |
| | Apply scale or questionnaire | It is used to apply the scale or questionnaire to a patient. The user can select any available questionnaire (e.g., UPDRS, PDQ-39, a personalized scale). |
| | Report generation | The user can visualize the scale or questionnaire scores obtained by a participant and by date of evaluation. |
| Data Management | Equipment control | This allows registering the type of equipment used in data collection. The user has to inform the name and detailed description of the equipment. |
| | Study control | This functionality is used to register information about the study of the research (e.g., description, start date, ethics approval information, number of sessions of data collection) |
| | Study group control | It is used to create a group of study, and it is necessary to inform the study, name, description, inclusion criteria, and exclusion criteria. This functionality is used to group participants of the study. |
| | Experimental protocol control | This allows for the creation and specification of experimental protocols. |
| | Experimental session control | This allows the user to store data resulting from an experimental session. Information such as study group, participant identification, equipment specification, protocol description, general observations, date and hour of the session, medication, file format (e.g., edf, csv, txt), and sharing permission are included. |
| | Data file download | This functionality allows the use of filters (e.g., participant, equipment, research group) to find and download files. |
| Idea collection and sharing | Unit supervisor control | The supervisor is responsible for the unit. This functionality allows recording basic information such as name, occupation, and contact. |
| | Unit control | It is used to register a sector of work in a hospital, clinic, or institution. The user has to inform the name, phone, and responsible for the sector. |
| | Idea management | This functionality allows adding an idea that can be converted into a project. It is essential to inform a description, keywords, a sector, and the identification of the person responsible for the idea. |
| Education and training | Category management | It is used to create categories to organize the symptom list in sections such as motor and non-motor symptoms, tutorials. |
| | Symptom control | The allows creating a record specifying a symptom, which includes title, detailed description, a link to a video sample, a link to extra information, and the category. |
| | List of Symptom | This lists symptoms or other kinds of information available in the system. |

| Module | Functionality | Description |
|---|---|---|
| Participant management | Set participant information | It controls the records of participants and patients. To insert a new volunteer is essential to fill in several fields such as medical record, institution, name, sex, birth date, weight, type of diagnosis, date of diagnosis, etc. |
| | Medication control | It is used to control different types of medication. It is essential to inform name, detailed description, dosage, unit. |
| | Medication association | This page is used to associate a participant with medication. To create an association, the user has to select the participant and the medication and fill in the dosage field. |
| Data query | Find data | This functionality finds records and files in the system. The user can choose some filters (e.g., type of data, age, sex, diagnosis, research group) to narrow the query, and the results show the files of data collection sessions, questionnaires, and scales presented in the entire system. |

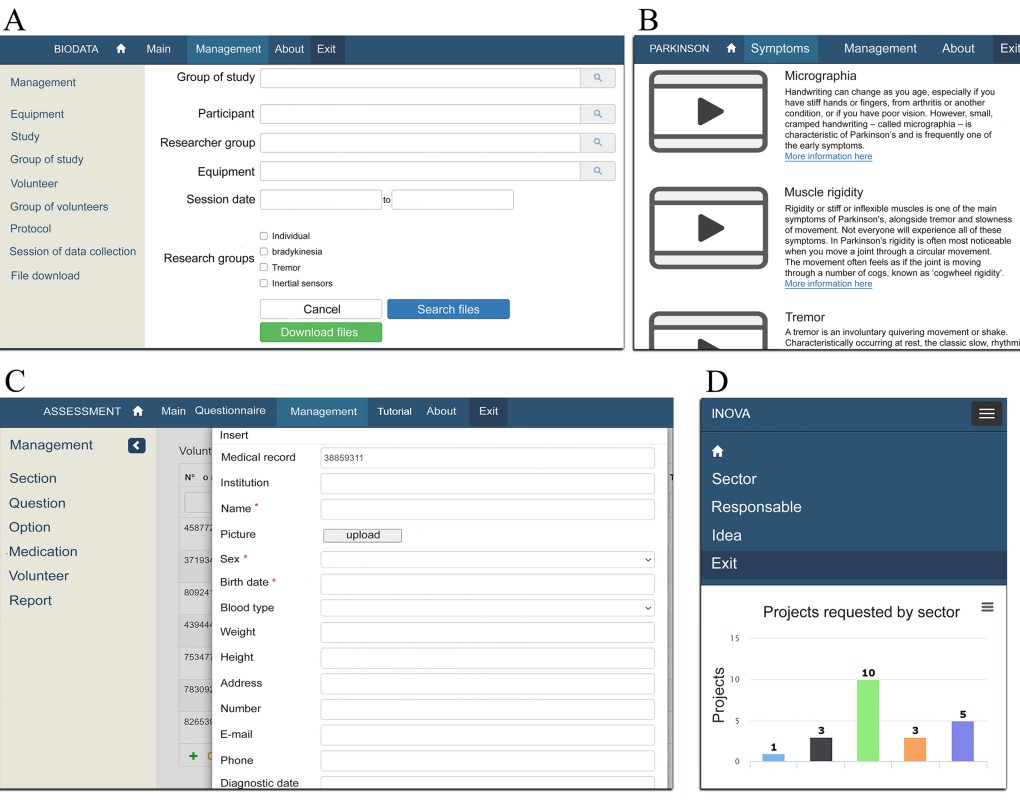

**Figure 4 Examples of screens developed in SIDABI.** (A–C) Respective examples of the screens implemented for the Data Management, Education and Training, and Clinical Assessment modules from the visualization in a PC; (D) illustrates the visualization of the Idea Collection and Sharing module from a smartphone.

Figure 4 exemplifies some functionalities implemented in the system. In (A) represents the screen for viewing and downloading data collection files, shared or not within research groups. The filters (e.g., protocol, equipment, session date, researcher group) are used to narrow the database's file search. Figure 4B illustrates the list of symptoms used to study and understand the theory and practical issues by videos of Parkinson's disease and

separated by categories. Figure 4C illustrates the volunteer registration screen with various specific information, and these data remain in common with other modules.

Finally, Fig. 4D illustrates the screen developed to view projects requested on the Idea Collection and Sharing module. This screen exemplifies the behavior of the responsiveness in using the system from a mobile device.

The SIDABI system presented so far has some characteristic similar to a clinical data warehouse (CDW) in which it is possible to storage distinct file format, a complex type of data, integration of a variety of applications, security, support decision making, support to researchers, helps in querying and presenting information, and data management. However, CDW has a complex infrastructure based on a variety of databases (raw data, meta data, summary data, data marts), operational data sources, a layer to Extract, Transform and Load (ETL) information, and many data access front end (e.g., analyzing, reporting, mining, medical information, business). Besides that, CDWs also deal with policies of data backups, the workload in the system, capacity of storage and no deletion of information (*Khalaf Hamoud, Salah Hashim & Akeel Awadh, 2018*).

Sistema Integrado de Dados Biomédicos focuses on the integration of web applications to take care of people with Parkinson's disease and help non-profit organizations. SIDABI is free and open-source, and in any necessity, programmers can change the code and the database related to medical record forms and functions that deal with information about Parkinson's disease and the system can cover other disorders. The entire structure and source code of SIDABI can be found in the Supplemental Information section.

## Usability and agreement results

The questionnaire that evaluates SIDABI was designed to obtain the examiner experience and the general characteristics of the system regarding the usability of the interface, error handling and system feedback messages, ease of use, satisfaction, and whether the system meets the tasks for which it was designed.

In this sense, from the 36 examiners, 83.33% had over five years of experience using computers, 77.78% were between 18 and 25 years old, and 52.78% used the computer for more than 10 h a week. The dataset and codes to compute these statistics can be found in the Supplemental Information section.

Figure 5 shows the answers obtained from the 36 examiners for each question of the SUS questionnaire. Most of the even-numbered questions yielded scores around 1–2 points, which characterizes a fair evaluation of negative purpose questions concerning the system's usability. On the other hand, most of the odd questions scored between 4 and 5, a high score for questions with a positive purpose, given the usability of the system.

The SUS scores are given in Fig. 6. The red dashed line represents the mean score obtained (82.99). The estimated standard deviation was 13.97, confirming the good usability of the system as the obtained SUS score is above the blue dotted line (68 points), which represents the acceptable value for a system evaluated with this scale.

In addition, it is possible to observe, according to the scale proposed by *Bangor, Kortum & Miller (2009)*, that the system is between good (71.4) and close to excellent

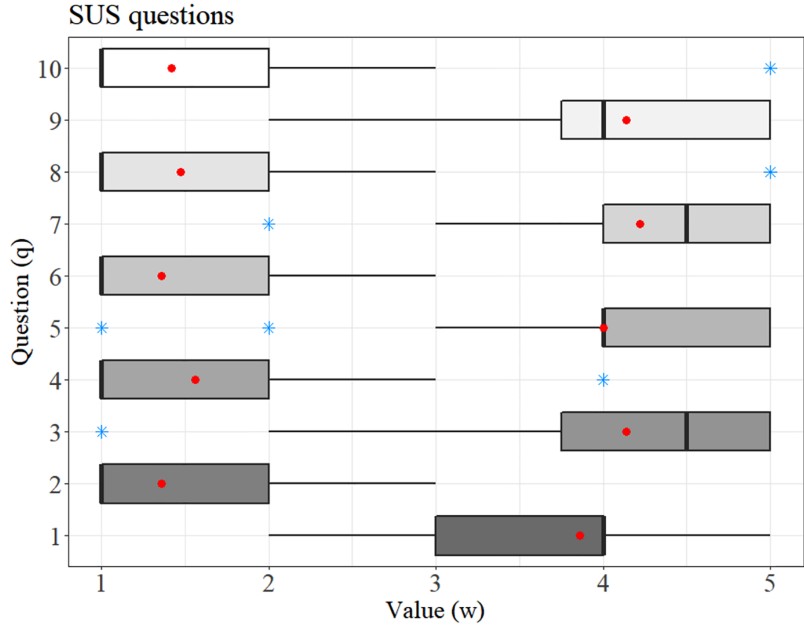

**Figure 5 Response distribution in which the average is highlighted by the red dot and the blue star represents the outliers.**

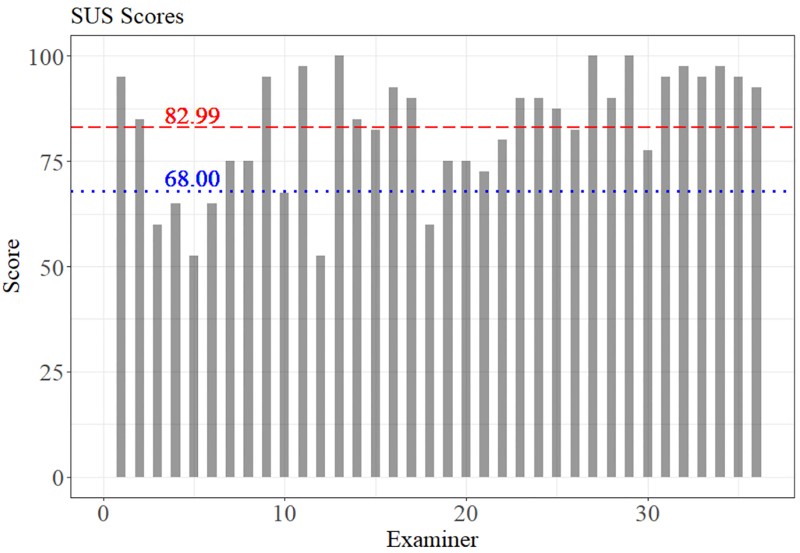

**Figure 6 SUS scores by examiners.** The value 68.00 represents the minimum average acceptable in the SUS score, and the value 82.99 was the mean score reached by the SIDABI.

(85.5 points). Most of the examiners gave scores higher than 82.99 points, and 7 of them evaluated the system with usability below 68 points.

The agreement between the examiners was verified using Kendall coefficient for the SUS scale score. The SIDABI reached 70.2% of concordance with $p = 5.9 \times 10^{-44}$, which suggests a good agreement among the evaluators.

## DISCUSSION

The proposal of an integrated system for managing data from patients with PD was presented, and the system could be installed in clinics that support people with PD, hospitals, and research laboratories. The project aims to improve data management, avoiding loss of datasets, fragmentation of information, decreasing research costs, and providing monitoring and clinical assessment of PD patients. Furthermore, SIDABI unifies education, research, and the caring of PD patients on the same system. All these functionalities in a free and open-source system can facilitate data management and help institutions with low resources or those that maintain the services through donations.

Additionally, the system improves the controlling of sensitive personal information, helping in the privacy rights defended by California Consumer Privacy Act (*Goldman, 2018*), Data Protection Law Enforcement Directive of European Union (*Quintel, 2018*) and General Personal Data Protection Act in Brazil (*Da Silva, Da Luz Scherf & Da Silva, 2020*). It is important to highlight that SIDABI assists with user authentication in data management, controls data manipulation, and helps the system administrator lock and unlock modules and permissions for functionality. In general, however, privacy rights (*Goldman, 2018*; *Quintel, 2018*; *Da Silva, Da Luz Scherf & Da Silva, 2020*) lay down rules governing the collection, handling, storage and sharing of personal data managed by organizations, prohibiting the sharing and use without consent of personal data. The system can also be downloaded, changed, and anyone can adequate the functionalities under specific necessities.

In the initial phase of the project, the system was installed on local servers of partners who use the system in hospitals, clinics, and in our research laboratory (NIATS). The system is currently installed on a dedicated local server in our laboratory. However, the ideal approach for a multiplatform web system is hosting on cloud services.

Despite that, the system is promising since several benefits are provided by the solutions developed. Diagnostic exams and assessments of specific PD symptoms, for instance, are easily stored and organized in SIDABI. In this context, the research that assesses cognitive dysfunction described by *Klinger et al. (2006)* could have their data stored in our system per patient, facilitating future analyzes, and allowing the comparison with other data. Likewise, the datasets resulting from the work of *Cunningham et al. (2009)* assessing hand dexterity, *Pastorino et al. (2013)* which record the patient's movement data to detect the ON/OFF medication condition, the evaluation and monitoring of gait proposed by the authors *Arango Paredes et al. (2015)*, *Pepa et al. (2015)* and *Patel et al. (2010)*; the monitoring of bradykinesia proposed by *Eskofier et al. (2016)*, and the speech assessment data described by the authors *Dimauro et al. (2017)*. All of these data could be collected from the same patient and stored in the same database and could be compared for a better understanding of the disease, to monitor the evolution of the symptoms, and to search for ways to provide better living conditions to patients in a much more effective way. In addition, any data can be shared by configuring groups of researchers, which facilitates collaboration and analysis of the information on the same platform.

*Cancela et al. (2013)* proposed a solution in integrating interinstitutional databases to facilitate collaborative information between the projects. However, our research goes further, with data stored in a structured way and with a secure organization to collect questionnaires, scales, exams and monitoring for each patient in an integrated manner and an easy-to-use interface.

The system architecture was designed to provide an easy-to-learn interface, with simplified functionalities and good satisfaction for the user who works on the platform. These characteristics can be observed in the usability evaluation applied to the system using the SUS scale, which obtained a good score (82.99 points) in which the agreement of the examiners was 70.2% ($p < 0.001$).

*Garzo et al. (2018)* applied SUS on its proposed system, and 37 people with PD evaluated the system with an average of 78.6 points on the final score. *Bank et al. (2018)* reached an average of 69.3 on SUS with 30 participants in the study (10 PD, 10 Stroke, 10 Healthy), and it was identified the need to improve the interaction of the participant hand opening with the virtual object. *Van der Meulen et al. (2016)* obtained an acceptable score on the SUS scale of 70.7 with 11 healthy participants on the evaluation of a game for motor assessment of PD patients. The study could identify problems within the wearable device used to reflect projected images and in the controller. *Kamdar & Wu (2016)* proposed a mobile application for monitoring mental health that reached an average of 74 points on the SUS scale with 13 healthy participants that reported a lack of desire to use the system frequently.

In addition, the authors *Wijaya, Munandar & Utaminingrum (2019)* obtained 51 points on the SUS scale with 20 evaluators, and despite the regular result, the test revealed the difficulty that users have in the system with the augmented reality marker. By contrast, *Devy, Wibirama & Santosa (2018)* reached an acceptable proposal with a SUS score of 75.5 points in the multimedia system for learning the English language, whose evaluation was carried out by 38 evaluators. Finally, *Indriana & Adzani (2017)* obtained a score on the SUS scale of 80.9 points, with 25 evaluators validating their e-commerce application. In this context, SIDABI obtained a good score (82.99) from its 36 examiners when compared to the studies presented.

Furthermore, in Fig. 5, question 2 (relating to the system's high complexity) and question 6 (representing the system inconsistency) reached an average between 1 and 1.5. These low values represent that the system is easy to use and has low inconsistency. Likewise, question 1 (reports satisfaction with using the system) and question 5 (evaluates the system's well-integrated functions) obtained an average between 3.7 and 4, a good score, but showing that these functionalities could be improved.

Finally, a good score was achieved regarding the user experience, a characteristic that influences quick learning when using the system. The user experience can be seen in question 7 (I think users will learn how to use the system quickly), which reached an average of 4.2, question 8 (evaluates if the system is confusing) reached an average of 1.5, and question 10 (represents the need to learn many things to be able to use the system) that obtained an average close to 1.5.

In the future, the module Data Query, which is illustrated in Fig. 2B, can be increased with more functionalities such as opening files of datasets providing statistical analysis to support general data understanding, some data mining tools, detection and classification of information.

The presented study had the intention to evaluate the usability of the system based on the knowledge and experience of professionals from distinct areas (i.e., researcher's labs, hospital, nonprofit organizations) aiming to understand whether the proposed system was adequate. Further researches could be carried out specifically with PD health professionals to evaluate the system usability, and also to the application of usability tests involving specialist programmers in usability. Even though SIDABI was built using good practices in usability and a set of responsive libraries to improve human-computer interaction, in future implementations, the standard IEC 62366, used in medical devices, could be adopted to improve usability. Another important study involving administrative gains in the use of SIDABI could be conducted. It will be possible to compare scenarios before and after the use of the system after a considerable number of institutions have adopted the system. The results of these studies could help to understand the average cost of direct spending on software for people with PD (Bovolenta et al., 2017).

The system described in this study has many segments that can be developed. The system is modular and was designed for future expansions. For future implementations, it is possible to do new implementations without a team interfering with each other's projects. Furthermore, the system is open-source: the source code is available to the communities to improve and make the functionalities grow up helping institutions that help people with PD.

## CONCLUSIONS

Sistema Integrado de Dados Biomédicos system is easy-to-use and has good potential in managing the biomedical data of people with Parkinson's disease. SIDABI was analyzed by experienced evaluators regarding the time spent using computer systems and obtained a good score (82.99 of 100 points) on the usability test. In this sense, the system can bring satisfaction to the user in the solution of their tasks.

Researches previously reported in the literature did not return results similar to this study until now. Therefore, the system is innovative in combining different solutions in this research area, collecting innovation in projects, learning and training about the disease, evaluating and monitoring patients with PD, and all of it on the same platform and with integrated functions. With an organized and secure database, it is possible to envision a future of savings in research, reducing time spent with recollect data, fragmentation of the information, and stimulating new projects, innovations and facilitating sharing data between them. Moreover, objectively enhance the decision-making of professionals regarding the symptoms of Parkinson's disease. The system is free and open-source to help institutions with low resources.

## ACKNOWLEDGEMENTS

We are thankful to the Parkinson Association of Triângulo (Associação de Parkinson do Triângulo, Uberlândia, Brazil) for their participation in this study.

### Funding

The present work was carried out with the support of the National Council for Scientific and Technological Development (CNPq), Coordination for the Improvement of Higher Education Personnel (Program CAPES/DINTER UFU—IFTM 225263/2014, CAPES—Program CAPES/DFATD-88887.159028/2017-00, Program CAPES/COFECUB-88881.370894/2019-01) and the Foundation for Research Support of the State of Minas Gerais (FAPEMIG-APQ-00942-17). Adriano de Oliveira Andrade, Adriano Alves Pereira and Marcus Fraga Vieira are a fellow of CNPq, Brazil (304818/2018-6, 310911/2017-6 and 306205/2017-3). The funders had no role in study design, data collection and analysis, decision to publish, or preparation of the manuscript.

### Grant Disclosures

The following grant information was disclosed by the authors:
National Council for Scientific and Technological Development (CNPq): CAPES/DINTER UFU—IFTM 225263/2014, CAPES/DFATD-88887.159028/2017-00, and CAPES/COFECUB-88881.370894/2019-01.
State of Minas Gerais: FAPEMIG-APQ-00942-17.
CNPq, Brazil: 304818/2018-6, 310911/2017-6 and 306205/2017-3.

### Competing Interests

The authors declare that they have no competing interests.

### Author Contributions

- João Paulo Folador conceived and designed the experiments, performed the experiments, performed the computation work, prepared figures and/or tables, authored or reviewed drafts of the paper, and approved the final draft.
- Marcus Fraga Vieira analyzed the data, authored or reviewed drafts of the paper, and approved the final draft.
- Adriano Alves Pereira analyzed the data, authored or reviewed drafts of the paper, and approved the final draft.
- Adriano de Oliveira Andrade conceived and designed the experiments, analyzed the data, authored or reviewed drafts of the paper, and approved the final draft.

### Ethics

The following information was supplied relating to ethical approvals (i.e., approving body and any reference numbers):

The research follows the Resolution 466/2012 of the National Health Council. The study was conducted at the Centre for Innovation and Technology Assessment in Health of the Federal University of Uberlândia (UFU), Brazil. The experimental protocol was approved by the Human Research Ethics Committee (CEP-UFU), CAAE Number: 93993118.4.0000.5152.

## Data Availability

The entire system source code, database, and basic instructions are available at Zenodo:

João Paulo Folador, & Adriano de Oliveira Andrade. (2020, October 29). Integrated Biomedical Data System (Version 3.1). Zenodo. DOI 10.5281/zenodo.4157489.

Files containing the system evaluation data by the 36 evaluators (not identified for ethical reasons) and the script developed in the R language are available at Zenodo:

Joao Paulo. (2020, October 30). jpfolador/susEvaluation: System usability evaluation (Version 1). Zenodo. DOI 10.5281/zenodo.4158836.

The database diagram is provided as a Supplemental File.

## Supplemental Information

Supplemental information for this article can be found online at http://dx.doi.org/10.7717/peerj-cs.396#supplemental-information.

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
