# Peer review of "Open-source data management system for Parkinson’s disease follow-up"

_PeerJ Computer Science, doi:10.7717/peerj-cs.396_

## Round 0.1 · original submission · Major Revisions

The authors have to respond to the reviewers. A point to point list should be provided.

·

Basic reporting

The level of language used is very adequate. Figures and references are accurate. However, the Rapid Application Development (RAD) concept is not trivial and could be better described (line 137). The article correctly describes the context of the study and helps to understand the issues related to the management of Parkinson's disease data. The examples given by the authors are relevant. The objectives of the study are correctly identified. The structure of the article is clear. The code of the application presented is open-source. The screenshots would perhaps benefit from more care.

Experimental design

The use of SUS questionnaires is an excellent introduction to assessing the usability of a tool, especially since the number of participants is very large. The technologies used are basic but well described.
The description of the development environment is not necessarily relevant, as the purpose of the paper is not a performance test.
The description of the VCM model seems superfluous in view of the paper issue. On the other hand, the choice of the MVC model compared to a MVVM model allowing more interactivity should be explained.
The evaluation method is very well described and well documented.

Validity of the findings

The results of the study are clearly stated. The results of the HMI are illustrated by screenshots. However, the Data Query module could be more detailed. The results obtained for the usability study are well described. The statistical method used is appropriate and puts the results into perspective. The subject of the study and the approach used are timely and relevant. It would be relevant for the author to situate the added value of SINOBI compared to a classical clinical data warehouse.
It would perhaps be interesting to specify if it is possible to extend the project to other pathologies or to explain the methodology envisaged for the integration of other data sources.

Reviewer 2 ·

Basic reporting

1. "Specific systems developed to manage people with PD and other neurological disorders has been previously reported. An effort to integrate interinstitucional databases can be found in [14], as well as applications using cell phone in [15], web-based systems in [15], [16], [17], and wearable sensors [16] to monitor PD patients, producing a large amount of data."

The studies cited in this paragraph could be better described, since they seem to be strong related to this one. Also, the four paragraphs before the mentioned one could be summarized, since the direct correlation with this study is not strong.

Experimental design

2. The authors opted to use SUS to assess usability and the support for that in the context of medical applications is weak. Also, just one cited study is related with medical application. It exist standards focused on medical applications, for instance, IEC 62366.

3. "Usability should aid the design of an interface that allows the user to perform tasks easily. In this way, the usability of a system should focus on developing interfaces that are easy to handle and quick to learn."

This definition is too much simple, and I agree with the importance of this part of the text to situate the reader and pace the reading. However, the definition of usability is not simple as that and should not be neglected in this kind of research paper.

4. "The usability test (SUS) of the system was performed by 36 examiners."
I advise to describe the examiners sample.

5. I did not noticed if a training was or was not offered to the examiners.

Validity of the findings

6. Who can be the user actor in Figure 3? What users in this case were considered by the authors?

7. It would be interested contrast the achieved results in terms of Usability with a profile more related with PD health professionals.

8. Why Kendall coefficient was chosen to evaluate the agreement among the evaluators? Make this clear in the text.

Additional comments

The paper presents a data management system for Parkinson's disease follow-up. The authors made the ideia itself very clear, and also there is no doubt about the importance of this kind of development to the field. Also, the manuscript is well written.

The usability was evaluated. What else could be evaluated in the proposed system considering its purpose?

---

## Round 0.2 · Minor Revisions

Minor revisions are still needed.

Reviewer 2 ·

Basic reporting

No comment.

Experimental design

No comment.

Validity of the findings

No comment.

Additional comments

All review points were addressed, explained and resolved.

Reviewer 3 ·

Basic reporting

English is adequate and the literary references too. The figures clarify important details of the text and the results are presented in a consistent manner. All terms were clear and objective.

Experimental design

The research subject is original and highly relevant. The necessary ethical aspects were used and the software will bring great gains for professionals and researchers linked to the disease in question. The state of the art was well done, even though there are not many works directly related to this software. This fact demonstrates how innovative this research is. The primary impacts were measured through the usability of the software and the rest will appear through its use in the academic community over time.

Validity of the findings

The tests, as well as the conclusion were done properly. The results are conclusive and the challenge is to present future works that can expand the areas of operation of this software.

Additional comments

a) Put the meaning of the acronym also in the native language: SIDABI.
b) Put information that involves financial aspects, related to the disease, in the item “backgound”. Example: how much does the Brazilian National Health System spend on Parkinson's annually?
c) Are there old databases for other diseases? What gains have they brought to the medical community? Could your software bring gains close to these? Note: Use more elements to value the work, especially numbers that involve financial aspects.
d) Why was MVC used? Do companies that develop complex software use it? What do they point to as strengths and weaknesses? Are there any success stories in the medical field documented? If so, they could be in the bibliographic references. The answers to these questions could be covered in a paragraph.
e) Why was an Apache server chosen? Does it meet all the needs of the project (security, availability, etc.)? Justify in the text.
f) Does the software respect the new rules of the “general data protection law” (Brazil)? Give details in the text.
g) The text should explain in detail the main "use cases". This would show the differential of the work for the Parkinson's treatment area and the complexity of the project. The contribution must be evidenced in all stages of the article.

---

## Round 0.3 · accepted · Accept

The authors have addressed the reviewers' comments.